# Discovery and Validation of Survival-Specific Genes in Papillary Renal Cell Carcinoma Using a Customized Next-Generation Sequencing Gene Panel

**DOI:** 10.3390/cancers16112006

**Published:** 2024-05-25

**Authors:** Jia Hwang, Seokhwan Bang, Moon Hyung Choi, Sung-Hoo Hong, Sae Woong Kim, Hye Eun Lee, Ji Hoon Yang, Un Sang Park, Yeong Jin Choi

**Affiliations:** 1Department of Hospital Pathology, Seoul St. Mary’s Hospital, College of Medicine, The Catholic University of Korea, 222 Banpo-daero, Seocho-gu, Seoul 06591, Republic of Korea; h202243050@cmcnu.or.kr (J.H.); dmsdl58@catholic.ac.kr (H.E.L.); 2Department of Urology, Seoul St. Mary’s Hospital, College of Medicine, The Catholic University of Korea, Seoul 06591, Republic of Korea; drvion@gmail.com (S.B.); toomey@catholic.ac.kr (S.-H.H.); ksw1227@catholic.ac.kr (S.W.K.); 3Department of Radiology, College of Medicine, Eunpyeong St. Mary’s Hospital, The Catholic University of Korea, Seoul 03312, Republic of Korea; cmh@catholic.ac.kr; 4Department of Computer Science and Engineering, Sogang University, Seoul 04107, Republic of Korea; yangjh@sogang.ac.kr (J.H.Y.); unsangpark@sogang.ac.kr (U.S.P.)

**Keywords:** papillary renal cell carcinoma, survival, gene, mutation, NGS, Korea

## Abstract

**Simple Summary:**

This study aimed to identify and validate genes specific for the survival of patients with papillary renal cell carcinoma (PRCC) in the TCGA-KIRP and Korean-KIRP databases. Using machine learning with statistical analysis, 40 survival-specific genes were identified in TCGA-KIRP. Of them, 10 were verified as survival-specific genes for Korean-KIRP patients through next-generation sequencing. Among these, BAP1, PCSK2, and SPATA13 showed significant survival specificity in both overall survival and disease-free survival. Survival gene signatures, including PCSK2, which are commonly obtained from 40 gene signatures in TCGA and 10 gene signatures in Korean databases, are expected to provide insight into predicting the survival of PRCC patients and developing new treatments.

**Abstract:**

Purpose: Papillary renal cell carcinoma (PRCC), the second most common kidney cancer, is morphologically, genetically, and molecularly heterogeneous with diverse clinical manifestations. Genetic variations of PRCC and their association with survival are not yet well-understood. This study aimed to identify and validate survival-specific genes in PRCC and explore their clinical utility. Materials and Methods: Using machine learning, 293 patients from the Cancer Genome Atlas-Kidney Renal Papillary Cell Carcinoma (TCGA-KIRP) database were analyzed to derive genes associated with survival. To validate these genes, DNAs were extracted from the tissues of 60 Korean PRCC patients. Next generation sequencing was conducted using a customized PRCC gene panel of 202 genes, including 171 survival-specific genes. Kaplan–Meier and Log-rank tests were used for survival analysis. Fisher’s exact test was performed to assess the clinical utility of variant genes. Results: A total of 40 survival-specific genes were identified in the TCGA-KIRP database through machine learning and statistical analysis. Of them, 10 (*BAP1*, *BRAF*, *CFDP1*, *EGFR*, *ITM2B*, *JAK1*, *NODAL*, *PCSK2*, *SPATA13*, and *SYT5*) were validated in the Korean-KIRP database. Among these survival gene signatures, three genes (*BAP1*, *PCSK2*, and *SPATA13*) showed survival specificity in both overall survival (OS) (*p* = 0.00004, *p* = 1.38 × 10^−7^, and *p* = 0.026, respectively) and disease-free survival (DFS) (*p* = 0.00002, *p* = 1.21 × 10^−7^, and *p* = 0.036, respectively). Notably, the PCSK2 mutation demonstrated survival specificity uniquely in both the TCGA-KIRP (OS: *p* = 0.010 and DFS: *p* = 0.301) and Korean-KIRP (OS: *p* = 1.38 × 10^−7^ and DFS: *p* = 1.21 × 10^−7^) databases. Conclusions: We discovered and verified genes specific for the survival of PRCC patients in the TCGA-KIRP and Korean-KIRP databases. The survival gene signature, including PCSK2 commonly obtained from the 40 gene signature of TCGA and the 10 gene signature of the Korean database, is expected to provide insight into predicting the survival of PRCC patients and developing new treatment.

## 1. Introduction

Papillary renal cell carcinoma (PRCC) accounts for 15–20% of all renal cell carcinomas (RCCs). It is the second most common kidney cancer after clear cell renal cell carcinoma (ccRCC) [1]. As reported in 2014, PRCC accounted for approximately 6.6% of all RCC cases in Korea [2]. PRCC is morphologically [3] and prognostically [1] heterogeneous, leading to varied clinical presentations on an individual basis. Sporadic PRCC occurs more commonly in patients with end-stage renal disease and acquired cystic disease [4]. However, the precise etiology remains elusive, implicating a combination of genetic, lifestyle, and environmental factors [5].

It is challenging to diagnose RCC early due to its asymptomatic nature, nonspecific symptoms, and an unpredictable growth pattern of tumors [6]. As the tumor progresses, treatment becomes increasingly difficult. Therefore, discovering biomarkers that can predict the progression and prognosis of RCC is essential. In particular, recognizing and studying genetic mutations associated with the survival of patients with RCC may facilitate personalized treatment. Additionally, it is worth noting that microRNAs have also been studied as potential biomarkers for tumor aggressiveness and metastasis in RCC [7].

Recent studies have uncovered prevalent mutations in PRCC. Type 1 PRCC commonly shows alterations in MET, whereas Type 2 PRCC frequently exhibits mutations in *CDKN2A* [1]. Furthermore, mutations in *TERT*, *FH*, *BAP1*, *SETD2*, *ARID2*, and genes associated with the Nrf2 pathway have been identified in patients with PRCC [1,8,9]. Unfortunately, research on the associations of these genetic mutations with the survival of PRCC patients is still insufficient. We have previously found associations between survival-specific gene mutations, including *CARD6*, through next generation sequencing (NGS) analysis of ccRCC patients [10]. This suggests the potential significance of survival-specific genes in predicting the survival outcomes of RCC patients.

This study aimed to discover the genes associated with the survival of 293 patients from the Cancer Genome Atlas-Kidney Renal Papillary Cell Carcinoma (TCGA-KIRP) database. It also aimed to determine whether these survival-related genes found in TCGA are also present in 60 Korean PRCC patients. In the TCGA-KIRP database, 44 (15%) were deceased, 54 (18.4%) experienced recurrence, and 12 (24.6%) had metastasis. In the Korean-KIRP database, 4 (12.4%) were deceased, 8 (13.3%) had recurrence, 8 (13.3%) had metastasis, and 8 (13.3%) experienced treatment failure after laparoscopic radial nephrectomy. We used machine learning (ML) to derive the genes associated with survival in PRCC, and next generation sequencing was conducted on 60 Korean PRCC patients using a customized PRCC gene panel consisting of 202 genes, including 171 survival-specific genes.

## 2. Materials and Methods

### 2.1. Ethical Statement

All procedures performed in this study were in accordance with the 1964 Helsinki declaration and its later amendments or comparable ethical standards. They were approved by the Institutional Review Board (IRB) of the Catholic University of Korea, Seoul St. Mary’s Hospital (IRB approval no. 2018-2550-0008, date of approval: 20 November 2018). This retrospective genetic study and treatment plans were conducted or developed according to clinical guidelines and standards of care. The results of the current genetic study did not affect the treatment plans of patients following surgery. Informed written consent was obtained from all patients.

### 2.2. Feature Selection and Machine Learning for Discovering Survival-Specific Genes in PRCC

In this study, ML was performed using Rapidminer version 7.3 (Boston, MA, USA) to identify mutated genes specific for PRCC survival in the TCGA database. From the TCGA-KIRP database we selected 293 patients who had non-silent and somatic mutations, along with obtaining their clinical information (accessed in August of 2017) [11]. The clinical data were obtained from the TCGA portal, and the Mutation Annotation Formatted file was obtained from UCSC Xena (UCSC, https://xenabrowser.net/) (accessed on 30 August 2017). There were 9646 genes in total. Variants were annotated using UCSC Xena [12]. The feature selection algorithm and classifiers used in the study are as follows: Information Gain, Chi-squared test, Minimum Redundancy Maximum Relevance (MRMR), Naïve Bayes, K-Nearest Neighbor, and Support Vector Machine. The performances of classification models in accordance with feature selections were analyzed. A 10-fold cross validation was used for model evaluation. The best-performing ML models were repeatedly utilized, and a total of 481 mutant genes related to six clinicopathological factors (age, sex, stage, recurrence, metastasis, and survival) were selected. To increase accuracy, only genes commonly found by the three methods were extracted. Of those 481 genes, 171 genes were statistically validated through analysis of variance (ANOVA) and Fisher’s exact test, considering their mutation frequencies, overall survival (OS), and disease-free survival (DFS).

### 2.3. Patients

Patients were enrolled from the Catholic University College of Medicine, Seoul, Republic of Korea. The inclusion criteria of this study comprised patients who underwent radical or partial nephrectomy, received a pathology diagnosis of PRCC, and expressed their willingness to participate by providing signed informed consent. All patients clinically diagnosed with a renal tumor underwent radical nephrectomy based on clinical indications. The pathology of each tumor was assessed by a pathologist specialized in renal cancer pathology. Only cases diagnosed as PRCC were included in the study. A total of 60 patients with PRCC were selected for this study. 

### 2.4. Samples

The biospecimens for this study were provided by the Biobank of Seoul St. Mary’s Hospital, the Catholic University of Korea. Formalin-fixed, paraffin-embedded (FFPE) samples of 20 normal-tumor paired tissues and 40 tumor-only tissues were obtained from PRCC patients (Korean-KIRP).

### 2.5. NGS Gene Panel Design for PRCC

A customized NGS gene panel for PRCC was designed with 202 genes. The panel consisted of 171 survival-specific genes, which were identified through ML in our research, 19 mutant genes (*BAP1*, *CUBN*, *DNAH8*, *FAT1*, *KDM6A*, *KIAA1109*, *KMT2C*, *KMT2D*, *LRP2*, *MET*, *MUC16*, *OBSCN*, *PBRM1*, *PCLO*, *PKHD1*, *SETD2*, *SYNE1*, *TTN*, and *WDFY3*) ranking above 3% in the TCGA-KIRP database, and 14 genes (*ALK*, *BRAF*, *BRCA1*, *BRCA2*, *EGFR*, *ERBB2*, *IDH1*, *IDH2*, *KIT*, *KRAS*, *MYC*, *MYCN*, *NRAS*, and *PDGFRA*) associated with solid tumors.

### 2.6. Targeted Library Preparation

Genomic DNAs were extracted from FFPE slides for library preparation. Genomic DNAs were fragmented (approximately 250 bp fragments) using the Bioruptor Pico Sonication System (Diagenode, Seraing, Belgium). They were processed for Illumina sequencing by end-repair, dA-tailing, adapter ligation, and pre-PCR for an indexed NGS library. The prepared gDNA library and capture probes were hybridized to capture target regions using a Celemics target enrichment kit (Celemics, Seoul, Republic of Korea). Customized capture probes were designed and chemically synthesized to hybridize target regions. Captured regions were further amplified by post-PCR to enrich the sample. The target-captured library was then sequenced on an Illumina NextSeq550 instrument (Illumina, San Diego, CA, USA) using a read layout of 2 × 150 bp.

### 2.7. Bioinformatics Analysis

Samples were sequenced with a Nextseq 550 platform (Illumina). BCL2FASTQ version 2.19.1.403 (Illumina) was used to demultiplex base-call image files into individual sequence read files (FASTQ format). All options and parameters followed default settings. Sequencing adapters were removed with AdapterRemoval version 2.2.2. [13] after low-quality bases were removed with an in-house code. All sequencing reads were aligned to the GRCh37 human genome with BWA-MEM (Burrows–Wheeler Aligner) version 0.7.17.software. This software uses the Burrows–Wheeler Transform algorithm to index the human genome sequence for calculating the constant complexity of each sequencing read. Post-align and recalibration processes were performed with Picard version 1.115 (http://broadinstitute.github.io/picard) (accessed on 15 January 2020) and GATK [14] version 4.0.4.0. Variant calling was performed with GATK Haplotype caller. All detailed parameters and options followed best practices.

### 2.8. Datasets

Data were gathered from the TCGA project. The TCGA-KIRP dataset encompassed clinical features from each patient, including demographics, tumor stage, vital status since the first surgical procedure, and corresponding sequencing reads from their cancer genome, regardless of the class of variant. This dataset can be found at https://www.cbioportal.org/study/summary?id=kirc_tcga (accessed on 7 April 2021). From the TCGA-KIRP dataset, 293 patients who had variants and clinical information were selected. The TCGA-KIRP dataset covered PRCC with the hg19 (GrCh37) annotation. The Korean-KIRP dataset contained clinical features from each patient, including demographics, tumor size, nuclear grade, tumor stage, tumor type, vital status since the first surgical procedure, and corresponding sequencing reads from their cancer genome, regardless of the class of variant. To determine the mutation frequency in the normal control population in South Korea, data from the Korea Biobank Array Project (referred to as KoreanChip) were also accessed after obtaining approval from the National Biobank of Korea, the Centers for Disease Control and Prevention, Republic of Korea (KBN-2019-019, approval date: 21 March 2019). The Korea Biobank Array Project was initiated in 2014 by the Korea National Institute of Health. It included 210,000 participants aged 40–69 years via the Korean Genome and Epidemiology Study [15] to implement a customized Korean genome structure-based array with high genomic coverage and abundant functional variants of low or rare frequency [16]. The KoreanChip comprised >833,000 markers, including >247,000 rare-frequency or functional variants estimated from >2500 sequencing data in Koreans. Of these >833,000 markers, 208,000 functional markers were genotyped. Particularly, >89,000 markers were present in East Asians. There was no common variant observed when comparing variants from PRCC patients to control (Korean Chip).

### 2.9. Data Pre-Processing

A variant call format file for storing gene sequence variations was processed using PLINK 1.9 (Shaun Purcell and Christopher Chang, www.cog-genomics.org/plink/1.9; accessed on 15 November 2021). Single nucleotide polymorphisms were genotyped to genomic variants using Ensembl Variant Effect Predictor (Version 96, Sarah E Hun (2019), http://apr2019.archive.ensembl.org/index.html; accessed on 15 November 2021). Identifiers for gene annotation were added using the biomaRt package from R [17]. To focus on the presence of cancerous mutations in PRCC, only variants identified in tumor tissues were considered. Non-synonymous mutations were only considered in this study after eliminating intron and synonymous variants. Variants with less than 2% of variant allele frequency, less than five alternate allele count, and less than 100 reading depth were excluded. Finally, benign and likely benign variants were discarded, as determined by the clinical significance of variants with reference to the ClinVar [18]. 

### 2.10. Gene Set Enrichment Analysis (GSEA)

Gene set enrichment analysis (GSEA) was performed using the Enrichr server [19] (https://maayanlab.cloud/Enrichr/) (accessed on 15 January 2024) to find out the biological processes and molecular functions of the survival-specific genes discovered from the TCGA-KIRP and Korean-KIRP databases. This involved performing Kyoto Encyclopedia of Genes and Genomes (KEGG) 2021 Human and Gene Ontology Biological Process 2021 databases. Significant molecular functions were selected with the following threshold: adjusted *p*-value < 0.05.

### 2.11. Statistical Analysis

To find the genes most closely associated with PRCC, Fisher’s exact test was performed to verify the strength of association between clinicopathological factors and survival or survival-specific genes. The analysis was performed by dividing subjects into two groups by age (<70 years vs. ≥70 years), nuclear grade (low grade (grades I and II) vs. high grade (grades III and IV)), tumor size (≤7 cm vs. >7 cm), and T stage (low stage (T1) vs. high stage (T2 and T3)). Fisher’s exact test was performed with the function of fisher.test in the Stats R-Package (RStudio Team (2016), Boston, MA, USA, http://www.rstudio.com, accessed on 15 October 2023). The survival probabilities of patients from the TCGA-KIRP and Korean-KIRP datasets were calculated with Kaplan–Meier curve and Log-Rank Test. These analyses were performed using Python libraries called Lifelines (0.26.4 version, Cameron Davidson-Pilon, https://lifelines.readthedocs.io/en/latest/) (accessed on 15 January 2024). All pairs of survival curves were compared using the Log-rank test module from lifeline statistics to confirm discrepancy between groups. Statistical significance was determined with *p* < 0.05 as a cutoff value (95% confidence level).

## 3. Results

### 3.1. Discovery of Survival-Specific Genes in TCGA-KIRP Database by Machine Learning

The workflow of this study is illustrated in Figure 1. To identify survival-specific genes in PRCC patients, we utilized data from the TCGA database. Table 1 summarizes the clinical findings of 293 patients from the TCGA-KIRP database. Of these 293 patients with PRCC, 214 (73.0%) were males and 78 (26.6%) were females, with males showing a 2.7-fold higher incidence. Of these 293 patients, 44 (15.0%) died, 54 (18.4%) experienced recurrence, and 12 (4.1%) developed metastasis.

To extract genes associated with survival from the TCGA-KIRP database, ML incorporating three feature selection methods (Information Gain, Chi-square, and MRMR) was applied to 9646 genes. Subsequently, 47 survival-related genes were identified. Survival analysis was performed for these 47 genes, revealing a statistically significant set of 40 survival-specific genes in OS or DFS. The genetic characteristics, metastasis status, and survival outcomes of patients with these 40 genes are detailed in Table 2. When closely observing these 40 survival-specific genes, the frequency of genetic mutations observed in these genes was found to be mutations with a low frequency, ranging from a minimum of 1 (0.34%) person to a maximum of 9 (3.07%) persons. Mutation types such as missense, truncating, in-frame, and splice were identified, of which missense and truncating mutations were commonly observed. Patients with mutations in those 40 survival-specific genes showed higher mortality rates than those without such mutations. Furthermore, among patients with metastasis, those with mutations in *ESRP2* (50%), *MUC17* (33%), *SNX7* (50%), and *SSX2IP* (100%) genes demonstrated a higher frequency of metastasis. In contrast, for the remaining genes the presence of mutations was either unrelated to metastasis or associated with equal or lower metastatic rates. 

Among those 40 survival-specific genes, 17 genes (*ACTR1B*, *BRD4*, *EPHB1*, *FBXW9*, *ITGA3*, *ITGA8*, *KPRP*, *MAOB*, *MUC17*, *MYH10*, *OR1S1*, *RAB40B*, *RTL1*, *SMARCA1*, *SSX2IP*, *TAS1R2*, and *THUMPD2*) exhibited specificity for both OS and DFS. Survival graphs for patients with these 17 genes are presented in Figure 2 and Figure 3. Patients with mutations in each of these genes consistently showed lower survival rates than those without such mutations. The collective survival graph for all 17 genes is illustrated in Figure 4. In the TCGA-KIRP dataset, 29 patients with mutations in any of the 17 genes displayed notably lower survival rates than 252 patients without mutations, showing statistical significance in both OS (*p* = 8.335 × 10^−33^) and DFS (*p* = 2.691 × 10^−24^). GSEA was performed for 17 survival-specific genes. A number of gene ontology molecular functions were revealed, and significantly enriched genes were found. *MAOB* gene showing primary amine oxidase activity (*p =* 0.00509, odds ratio (OR): 249.72) and oxidoreductase activity (*p =* 0.005936, OR: 208.09) and *BRD4* showing histone reader activity (*p* = 0.006781, OR: 178.36) and tRNA (Guanine) methyltransferase activity (*p* = 0.009313, OR: 124.83) were related to *THUMPD2* genes (Appendix A).

The clinical information and results of survival-specific genes for 12 patients with PRCC metastasis in 293 patients of TCGA-KIRP are detailed in Table 3. The mean age of these 12 patients with metastasis was 55.3 years. Among them, one patient (8.3%) was aged 30–39 years, three (25.0%) were aged 40–49 years, two (16.7%) were aged 50–59 years, five (41.7%) were aged 60–69 years, and one (8.3%) was aged 70–79 years. Of these 12 patients with metastasis, 11 (91.7%) were males and 1 (8.3%) was female, with the majority of patients with metastasis being males. The results of TNM staging showed stages T2 or higher, with four (33.3%) patients classified as T2, seven (58.3%) patients as T3, and one (8.3%) patient as T4. Lymph node metastasis was identified in eight (66.7%) patients. Of the 12 patients with metastasis, 7 (58.3%) died. Tumor type information was available for three (25.0%) patients, with all three being classified as Type II. Upon investigating the presence of survival-specific genes in the 12 patients with metastasis, it was found that mutant genes were only observed in 7 patients, who died. Among the 40 survival-specific genes, three (25.0%) patients had mutations in *MUC17*, two (16.7%) in *SSX2IP*, and one (8.3%) patient each in the remaining nine genes (*CIITA*, *ESRP2*, *MAOB*, *MYH10*, *PPM1F*, *RTL1*, *RYR1*, *SNX7*, and *THUMPD2*).

#### 3.1.1. Verification of Survival-Specific Genes in Korean-KIRP Patients through NGS Analysis

The clinical and pathological findings of the 60 patients in the Korean-KIRP cohort are summarized in Table 4. The mean age of these patients was 64 years, with 11 (18.3%) patients in the age group of 40–49 years, 11 (18.3%) in 50–59 years, 15 (25%) in 60–69 years, 15 (25%) in 70–79 years, 7 (11.7%) in 80–89 years, and 1 (1.7%) in 90–99 years. Of these 60 patients, 45 (75%) were males and 15 (25%) were females, with males showing a three-fold higher incidence. Tumor types included Type I in 15 (25%) patients and Type II in 45 (75%) patients, with Type II showing a three-fold higher ratio. Nuclear grades included Grade II in 25 (41.7%) patients, Grade III in 34 (56.7%) patients, and Grade IV in 1 (1.7%) patient. Tumor sizes were categorized as ≤7.0 cm in 52 (86.7%) patients and >7.0 cm in 8 (13.3%) patients. Among patients with tumor sizes ≤7.0 cm, one (1.9%) died. Among those with tumor sizes >7.0 cm, three (37.5%) died. There was a significant association between tumor size and mortality (*p* = 0.006). Regarding TNM classification for PRCC, T stage was T1 for 47 (78.3%) patients, T2 for 5 (8.3%) patients, and T3 for 8 (13.3%) patients. N stage was N1 for three (5%) patients. M stage was M1 for eight (13.3%) patients. Of 47 patients with a low stage (T1), one (2.1%) died. Of 13 patients with a high stage (T2 and T3), 3 (23.1%) died. There was a significant correlation between T stage and mortality (*p =* 0.029). Of 52 patients without metastasis in the M stage, there were no deaths. Of eight patients with metastasis, four (50%) died. There was a significant association between M stage and mortality (*p* = 0.0001). Of 52 patients without recurrence, there were no deaths. Of eight (13.3%) patients with recurrence, four (50.0%) died (*p* = 0.0001). Laparoscopic radical nephrectomy (LRN) was performed for all patients. Of 52 patients without evidence of the disease after treatment, none died. However, metastasis occurred in eight (13.3%) patients for whom treatment failed. Of these eight patients, four (50.0%) died of renal cell carcinoma (*p* = 0.0001).

In 60 Korean-KIRP patients, mutations were detected in 177 out of 202 genes analyzed using the NGS_PRCC gene panel. Of these 177 genes, 10 genes (*BAP1*, *BRAF*, *CFDP1*, *EGFR*, *ITM2B*, *JAK1*, *NODAL*, *PCSK2*, *SPATA13*, and *SYT5*) showed statistically significant survival specificity in OS or DFS. The gene mutations and survival analysis results for these 10 survival-specific genes are presented in Table 5. The most commonly observed mutations in the 10 survival-specific genes within the Korean-KIRP cohort were missense and in-frame mutations. Patients with mutations in *BAP1* (100%) and *PCSK2* (100%) among the 10 survival-specific genes all died. Among the five genes associated with high metastasis rates, patients with mutations in *BAP1* (100%), *ITM2B* (100%), *NODAL* (100%), and *PCSK2* (100%) had metastasis to distant organs, and patients with *JAK1* mutations showed a higher metastasis rate (67%) than those without this mutation. Five genes (*BAP1*, *BRAF*, *EGFR*, *PCSK2*, and *SPATA13*) showed survival specificity in OS (*p* = 0.00004, *p* = 0.034, *p* = 0.034, *p* = 1.38 × 10^−7^, and *p* = 0.026, respectively) and eight genes (*BAP1*, *CFDP1*, *ITM2B*, *JAK1*, *NODAL*, *PCSK2*, *SPATA13*, and *SYT5*) exhibited survival specificity in DFS (*p* = 0.00002, *p* = 0.004, *p* = 0.027, *p* = 0.0004, *p* = 0.002, *p* = 1.21 × 10^−7^, *p* = 0.036, and *p* = 0.021, respectively). The genes *BAP1*, *PCSK2*, and *SPATA13* demonstrated survival specificity in both OS and DFS. Through GSEA for the 10 gene signatures, gene ontology molecular functions were revealed. Particularly, *EGFR* and *JAK1* showed significantly enriched functions, including protein tyrosine kinase activity (*p* = 0.0167), protein phosphatase binding (*p* = 0.0167), and ubiquitin protein ligase binding (*p* = 0.03922). *JAK1* also exhibited associations with functions such as growth hormone receptor binding (*p* = 0.02796) and non-membrane spanning protein tyrosine kinase activity (*p* = 0.04588) (Appendix A). 

Table 6 shows clinical information for eight patients with metastasis to distant organs among the 60 PRCC patients and details of the survival-specific genes found in these individuals. The mean age of the eight metastatic patients was 64.5 years, with three (37.5%) patients in the age group of 40–49 years, one (12.5%) patient in 50–59 years, one (12.5%) patient in 70–79 years, two (25%) patients in 80–89 years, and one (12.5%) patient in 90–99 years. Of eight patients, five (62.5%) were males and three (37.5%) were females, with males showing a roughly two-fold higher ratio. All (100%) eight patients had Type II tumors. Nuclear grade was III in seven (87.5%) patients and IV in one (12.5%) patient. Regarding T stage in the TNM classification, there were two (25%) patients at the T1 stage, two (25%) at the T2 stage, and four (50%) at the T3 stage. Lymph node metastasis was present in two (25%) patients. Among the eight patients with metastasis, four (50%) died. All eight patients experienced metastasis due to treatment failure following laparoscopic radical nephrectomy. Lung metastasis was observed in six (75%) out of eight patients. Additionally, metastases were found in the lymph nodes in two (25%) patients, adrenal gland in one (12.5%) patient, bone in one (12.5%) patient, liver in one (12.5%) patient, and vagina in one (12.5%) patient. Analysis of the 10 survival-specific genes identified in the eight patients with metastasis revealed that *CFDP1* was present in four (50%) patients, *SPATA13* was present in three (37.5%) patients, and *JAK1* was present in two (25%) patients. The remaining seven genes (*BAP1*, *BRAF*, *EGFR*, *ITM2B*, *NODAL*, *PCSK2*, and *SYT5*) were present in one patient (12.5%) each.

The OS and DFS graphs for the 10 survival-specific genes are shown in Figure 5. In most cases, patients with gene mutations had shorter OS and DFS survival rates than those without mutations. However, patients with mutations in the *ITM2B*, *JAK1*, *NODAL*, and *SYT5* genes showed longer OS. The collective survival graph of *BAP1*, *PCSK2*, and *SPATA13* genes that demonstrated survival specificity in both OS and DFS is presented in Figure 6. The 9 patients with mutations in at least one of these three survival-specific genes exhibited shorter survival rates than the 51 patients without these mutations, showing statistical significance in both OS (*p* = 0.026) and DFS (*p* = 0.036).

#### 3.1.2. A Survival-Specific Gene Commonly Identified in Both TCGA-KIRP and Korean-KIRP Databases

The only survival-specific gene commonly identified in the TCGA-KIRP and Korean-KIRP databases was *PCSK2*. The analysis results for *PCSK2* are summarized in Table 7. In the TCGA-KIRP database, *PCSK2* gene mutations included a missense variant (c.850C>T; p.Leu284Phe) and an in-frame insertion variant (c.10_12dupGGT; p.Gly4dup). In the Korean-KIRP cohort, a *PCSK2* gene mutation included a missense variant (c.1879G>T; p.Val627Leu). In th eTCGA-KIRP dataset, *PCSK2* mutations were detected in two (0.68%) patients. One of them, a 75-year-old male diagnosed with Type 2 PRCC, had a TNM classification of T1bN1MX. The patient had a recurrence after 9.7 months and died after a follow-up period of 13 months. Another patient, a 74-year-old male diagnosed with Type 1 PRCC, had a TNM classification of T2N0M0 and died 76 months after the follow-up period. In the Korean-KIRP dataset, one (1.67%) patient with a *PCSK2* mutation, a 93-year-old female diagnosed with Type 2 PRCC (T3N1M1), experienced recurrence 21 days after surgery and died after a follow-up period of 11 months. Patients with *PCSK2* gene mutations showed shorter survival periods in both OS and DFS, consistently observed in both the TCGA-KIRP and Korean-KIRP databases (Figure 7). In the TCGA-KIRP database, among the two patients with *PCSK2* gene mutations, both (100%) died and one (50%) experienced recurrence (lymph node metastasis). In contrast, among the 279 patients without *PCSK2* gene mutations, 39 (13.9%) died and 48 (17.2%) experienced recurrence, indicating lower survival rates in patients with mutations, including OS (*p* = 0.010) and DFS (*p* = 0.301). In addition, in the Korean-KIRP dataset, one patient with a *PCSK2* gene mutation (100%) experienced both recurrence and death. Among the 59 patients without the mutation, 3 (5.1%) patients died and 7 (11.9%) experienced recurrence. Consequently, Korean patients with *PCSK2* gene mutations demonstrated lower survival rates, showing significantly lower OS (*p* = 1.38 × 10^−7^) and DFS (*p* = 1.21 × 10^−7^) than those without those mutations.

#### 3.1.3. Clinicopathological Significance of Survival-Specific Genes in Korean-KIRP

We investigated mutations in 177 genes detected in 60 Korean KIRP patients with the NGS_PRCC panel and analyzed their association with clinical factors. Statistically significant genes are summarized in Appendix A. In particular, among genes showing clinicopathological relevance, three genes (*CFDP1*, *JAK1*, and *SPATA13*) were identified as survival-specific genes (Table 8). *CFDP1*, one of the 10 survival gene signatures, demonstrated additional statistical significance in the M stage, recurrence, and response to LRN (*p* = 0.013 for all) besides survival. *JAK1* also exhibited statistical significance in the M stage, recurrence, and response to LRN (*p* = 0.044 for all). Lastly, *SPATA13* showed statistical significance in tumor size (*p* = 0.013) and T stage (*p* = 0.018). Among the three genes showing clinicopathological associations in addition to survival in the Korean-KIRC dataset, *JAK1* demonstrated statistical significance in recurrence (*p* = 0.030) in the TCGA-KIRP dataset, consistent with results observed in the Korean-KIRP dataset. *PCSK2*, identified as a survival-specific gene in both cohorts, did not demonstrate associations with other clinicopathological factors in the Korean-KIRP cohort. In contrast, in the TCGA-KIRP cohort it showed an association with death (*p* = 0.021) (Appendix A).

## 4. Discussion

The accumulation of molecular findings for RCC is causing major changes in pathological diagnosis. It has especially accelerated the expansion of molecular classification through the revision of the WHO classification in 2022 [20]. Recently, the growing interest in genetic mutations influencing the prognosis and treatment of RCC has prompted studies utilizing the genetic mutations [21,22] and RNAseq data [23,24] available in open databases such as TCGA. 

Lasorsa et al. highlighted the importance of cancer stem cell signaling pathways such as Notch, Wnt, and Hedgehog in ccRCC, and they proposed a treatment strategy targeting cancer stem cells, which are presumed to cause cancer recurrence and metastasis [25]. However, as of now, studies on genetic mutations associated with survival in PRCC are still lacking.

PRCC generally demonstrates a more favorable prognosis compared to ccRCC [26]. Genetic variations related to histologic subtypes have also been reported [1]. Common mutations, such as MET alterations in Type 1 and CDKN2A in Type 2 PRCC, have been consistently observed in current studies. Additionally, various mutations, including TERT, FH, BAP1, SETD2, ARID2, and Nrf2 pathway genes, have been identified [1,8,9].

We identified 40 survival-specific genes from 293 PRCC patients (TCGA-KIRP) through ML and statistical validation. We particularly emphasized the clinical utility of 17 survival-specific gene signatures that showed significance in both OS and DFS. Additionally, through collective survival analysis, we confirmed the ability to predict the survival of PRCC patient based on the presence or absence of gene mutations.

There were some difficulties in discussing our findings because many of the key genes we discovered had not previously been reported for survival specificity in RCC. Therefore, below we discuss the differences from the data reported to date for the above seven genes. However, even in cases where relevant data were available, there were limitations in discussing direct correlations with our PRCC results because most of them were based on ccRCC rather than PRCC.

The Bromodomain Containing 4 (*BRD4*) gene is known to play various roles in transcriptional regulation by RNA polymerase II [27]. Through our research, we discovered its survival specificity in OS (*p* = 0.020) and DFS (*p* = 0.009). The Class II Major Histocompatibility Complex Transactivator (*CIITA*) gene is one of the Human Leukocyte Antigen (HLA) class II regulatory genes that can induce the expression of immune system genes [22]. Its survival specificity in OS (*p* = 0.000002) was also revealed. Although studies associating *BRD4* and *CIITA* genes with PRCC patient survival have not yet been reported, previous studies on CCRCC patients have reported that overexpression of *BRD4* (*p* = 0.0003) and *CIITA* (*p* = 0.037) genes is associated with lower survival rates [28,29]. Notably, in breast cancer constitutive CIITA expression was observed in poorly metastatic cases, indicating a correlation between gene expression and metastasis [30].

The CD1c Molecule (*CD1C*) gene encodes a protein presenting Class I MHC antigens to T cells, crucial in immune disorders [31]. In this study, *CD1C* showed survival specificity in OS (*p* = 0.017). The Cytochrome P450 Family 51 Subfamily A Member 1 (*CYP51A1*) gene catalyzes a key step in cholesterol biosynthesis. In our analysis of the TCGA-KIRP database, *CYP51A1* showed survival specificity (OS, *p* = 0.001). Although there have been no reports associating *CD1C* or *CYP51A1* gene mutations with the survival of PRCC patients, a previous study on CCRCC patients has reported that high expression of *CD1C* gene is associated with higher survival rates (*p* < 0.0001) [32].

The Collagen Type V Alpha 1 Chain (*COL5A1*) gene is implicated in ccRCC metastasis [33]. In patients with CCRCC, expression of the *COL5A1* gene has been associated with OS (HR: 1.876; *p* = 0.027) and recurrence-free survival (HR: 4.751; *p* < 0.001) [33]. Furthermore, a 10-gene (*COL1A1*, *COL5A1*, *COL11A1*, *FN1*, *ICAM1*, *ITGAL*, *ITGAM*, *ITGB2*, *THBS2*, and *TIMP1*) expression signature containing *COL5A1* observed in CCRCC patients has been reported to be associated with poor survival (HR: 2.85, *p* = 5.7 × 10^−10^) [34]. Particularly in metastatic RCC patients, the expression of *COL5A1* has been reported to be associated with lower survival rates [35]. While there have been no reports on *COL5A1* in PRCC patients, our study identified its survival specificity in OS (*p* = 0.001) among PRCC patients.

The F-Box and WD Repeat Domain Containing 9 (*FBXW9*) gene, located on 19p13.13, belongs to the WD40 repeat-containing F-box proteins (FBXWs) family. *FBXWs* function as E3 ubiquitin ligases, mediating protease-dependent protein degradation. Unlike other FBXWs members, the function and substrates of *FBXW9* have not yet been studied in human diseases [36]. The expression of *FBXW9* in pan-cancer and normal tissues was found to be increased in tumor tissues compared to that in normal tissues of PRCC patients. High expression of *FBXW9* was associated with good OS (HR = 0.36, *p* = 0.0046) in PRCC [36,37] and poor OS in breast and bladder cancers [37]. In the TCGA-KIRP data we analyzed, *FBXW9* mutations were survival specific for both OS (*p* = 0.0001) and DFS (*p =* 0.00001). However, studies on an association between *FBXW9* mutations and survival in PRCC patients have not yet been reported.

The Integrin Subunit Alpha 8 (*ITGA8*) gene encodes the alpha 8 subunit of the heterodimeric integrin alpha8beta1, which is closely associated with the development and progression of tumors [38]. In our study, *ITGA8* showed survival specificity in both OS (*p* = 3.87 × 10^−7^) and DFS (*p* = 1.42 × 10^−7^). However, an association between *ITGA8* and survival of PRCC patients has not yet been reported. In contrast, in a study on ccRCC patients, low expression of the *ITGA8* gene was found to be associated with lower survival rates (*p* < 0.0001) [39].

Among the above seven genes, studies on the association with survival in PRCC were reported only on the FBXW9 gene, while the results of the remaining six genes were obtained in ccRCC studies. Moreover, these studies analyzed the relationship between gene expression and survival, and they did not study the relationship with genetic mutations as in our study. Overexpression of FBXW9 has been reported to be associated with good OS in PRCC [36,37], and in our study cases with genetic mutations showed a worse prognosis in both OS and DFS. These results suggest that they may be similar to each other. The outcomes reported in patients with CCRCC showed that overexpression of the BRD4, CIITA, and COL5A1 genes was associated with lower survival [28,29] and that overexpression of the genes CD1C and ITGA8 was associated with higher survival [32,39]. In particular, it has been reported that the expression of COL5A1 in metastatic ccRCC patients is associated with poor survival [33,34].

In this study, the authors designed a customized PRCC gene panel for validating identified genes and conducted NGS analysis for 202 genes in 60 Korean-KIRP patients. As a result, 10 gene signatures among 177 gene mutations detected in Korean PRCC were found to be survival-specific in the Korean population. The association with survival of all 10 survival-specific genes (*BAP1*, *BRAF*, *CFDP1*, *EGFR*, *ITM2B*, *JAK1*, *NODAL*, *PCSK2*, *SPATA13*, and *SYT5*) discovered in this study was discussed. However, there has been no report on the association with survival in RCC for nine genes, except for the *NODAL*. Just as most of the genes discovered in the TCGA-KIRP dataset had not been reported to be related to survival in RCC, the same was true for the genes discovered in the Korean-KIRP dataset.

Interestingly, among these survival-specific genes, only *PCSK2* was common between the Korean-KIRP and TCGA-KIRP databases. In both the Korean-KIRP and TCGA-KIRP databases, *PCSK2* demonstrated an association with survival, including OS (*p* = 1.38 × 10^−7^ and *p* = 0.010, respectively) and DFS (*p* = 1.21 × 10^−7^ and *p* = 0.301, respectively).

Proprotein Convertase Subtilisin/Kexin Type 2 (*PCSK2*) encodes a protein-degrading enzyme responsible for activating inactive prohormones into active peptides. As of the current RCC research, no reports exist on the association between *PCSK2* mutations and survival. In some previous esophageal cancer studies, Ma and Luo reported that when creating a gene signature with differential expression genes, *PCSK2* can negatively influence the prognosis of esophageal squamous cell carcinoma patients (OS: *p* = 0.0044, DFS: *p* = 0.100) [21]. Additionally, Li et al. [40] reported that *PCSK2* is associated with the survival of esophageal cancer patients as an immune-related gene (HR: 1.30, *p* = 0.034). As a result of this study, the PCSK2 gene mutation was observed at a low frequency of 0.68% and 1.67%, respectively, in both TCGA and Korean PRCC patients. As research on this gene is still insufficient, it is believed that more interest and additional research are needed. In addition to *PCSK2*, *BAP1* and *SPATA13* were among those survival-specific genes identified in the Korean-KIRP dataset that exhibited survival specificity in both OS and DFS. 

BRCA1 associated protein-1 (*BAP1*) is a gene located on 3p21. It has been reported to be a driver gene in ccRCC [41] and PRCC [9]. In ccRCC, survival specificity based on genetic variations has been confirmed [42,43,44]. Pan-Kidney Cancer analysis for 843 RCC patients has revealed that *BAP1* mutations are associated with lower survival rates in the ccRCC group, but not in PRCC or ChRCC [45]. Our previous study on Korean ccRCC patients also reported an association between *BAP1* mutations and lower DFS (*p* = 0.029, r = −0.465) [10]. In this study, the survival specificity of *BAP1* showed differences depending on the cohort. In the Korean-KIRP dataset, it was significant in both OS (*p* = 0.00004) and DFS (*p* = 0.00002), whereas it was not significant in OS or DFS in the TCGA-KIRP dataset. It is suspected that the differences in the survival specificity of *BAP1* mutations among PRCC cohorts may be due to racial differences, but additional studies in large groups of Asians are required to confirm this. There is currently no report on the survival specificity of *BAP1* mutations in Asian PRCC cohorts.

Spermatogenesis-Associated Protein 13 (*SPATA13*) contributes to the regulation of cell migration and adhesion [46]. In our study, the mutation frequency of *SPATA13* was notably high at 15%. It demonstrated a survival specificity in both OS (*p* = 0.026) and DFS (*p* = 0.036). However, the association between *SPATA13* mutations and survival has not been reported in any cancer, including RCC. In the Korean-KIRP dataset, *SPATA13* mutations were additionally associated with tumor size (*p* = 0.013) and T stage (*p* = 0.018). However, the association between *SPATA13* and tumor size or T stage in cancer has not yet been reported. The authors conducted collective survival analysis using three survival-specific genes (*BAP1*, *PCSK2*, and *SPATA13*) identified in the Korean-KIRP database, demonstrating the ability to predict the survival of PRCC patients based on the presence of genetic mutations. In the Korean-KIRP dataset, the genes showing survival specificity in OS were *BRAF* and *EGFR*, whereas the genes exhibiting specificity in DFS were *CFDP1*, *ITM2B*, *JAK1*, *NODAL*, and *SYT5*.

The B-Raf Proto-Oncogene, Serine/Threonine Kinase (*BRAF*) gene encodes a protein belonging to the serine/threonine protein kinases RAF family known to induce abnormal cell proliferation when overexpressed or mutated [47]. It has been reported that such *BRAF* gene mutations are very rare in RCC [48]. In our study, *BRAF* mutations were rare, with frequencies of 3% in the TCGA-KIRP dataset and 5% in the Korean-KIRP dataset. In our analysis of the Korean-KIRP dataset, *BRAF* mutations were specifically associated with OS (*p* = 0.034). This finding aligned with another study involving Korean RCC patients, where *BRAF* mutations were associated with shorter OS in 56 metastatic RCC patients (Log-rank *p* < 0.001, hazard ratio (HR): 4.259) [49].

Epidermal growth factor receptor (*EGFR*) encodes a tyrosine kinase receptor of the ErbB family. It is often abnormally activated in epithelial tumors [50]. In our study, *EGFR* mutations were rare, with frequencies of 1% in the TCGA-KIRP dataset and 5% in the Korean-KIRP dataset. *EGFR* mutations in Korean-KIRP patients were significantly associated with OS (*p =* 0.034) in our study. However, reports on the association between the *EGFR* mutation and survival in PRCC are lacking. Conversely, in ccRCC, *EGFR* overexpression has been linked to adverse clinical outcomes. Studies have reported that *EGFR* overexpression is associated with a higher nuclear grade (*p* < 0.001), a larger tumor size (*p* = 0.011), and a shorter patient survival (*p* = 0.046) [51]. Moreover, the expression of *EGFR* in ccRCC patients with metastasis was higher [52]. PRCCs typically show limited response to cytokine therapy and a modest response to TKIs [53]. Treatment options for advanced PRCC patients are constrained and largely extrapolated from ccRCC treatment, with clinical trial enrollment recommended [53].

Cranio Facial Development Protein 1 (*CFDP1*) contributes to cell cycle progression. In our Korean-KIRP database, the gene mutation frequency was high at 15%, showing specificity for DFS (*p =* 0.004). However, there have been no reports on the association between *CFDP1* mutations and survival in RCC. In contrast, in hepatocellular carcinoma, *CFDP1* overexpression was associated with shorter OS (hazard ratio: 2, *p* = 0.012) and DFS (hazard ratio: 1.8, *p* = 0.0099), promoting cancer progression by activating the NEDD4/PTEN/PI3K/AKT signaling pathway [54]. Additionally, in neuroblastoma, *CFDP1* overexpression has been reported to be associated with shorter OS (*p* = 9.7 × 10^−9^) and event-free survival (*p* = 5.2 × 10^−7^) [55]. Among the eight Korean PRCC patients in whom metastasis was discovered in our study, the survival-specific genes observed at high frequency were *CFDP1* in four patients (50%), *SPATA13* in three patients (37.5%), and *JAK1* in two patients (25%).

Janus Kinase 1 (*JAK1*) plays a crucial role in cancer progression [56]. In our Korean-KIRP study, *JAK1* mutations were found at a frequency of 5%, negatively impacting DFS (*p* = 0.0004) and showing associations with M stage, recurrence, and response to LRN (*p* = 0.044 for each). Additionally, *JAK1* exhibited an association with recurrence in the TCGA-KIRP cohort (*p =* 0.030). While other ccRCC studies have reported an upregulation of *JAK1* in renal cell carcinoma (*p* < 0.001) [57], specific survival associations with this gene in RCC have not been reported to date. On the contrary, a breast cancer study by Chen et al. [58] indicated that overexpression of *JAK1* is associated with a favorable prognosis (OS, HR: 0.52, 95% CI: 0.42–0.65, *p* = 1.5 × 10^−9^).

Nodal Growth Differentiation Factor (*NODAL*) is a TGF-b–related embryonic morphogen expressed in various human cancers, acting as a master regulator of tumor cell plasticity and tumorigenicity. Due to its absence in most normal adult tissues and overexpression in aggressive tumor cells, *NODAL* is considered a potentially valuable therapeutic target [59]. In the Korean-KIRP cohort, *NODAL* mutations were identified at a frequency of 1.67%, showing specificity for DFS (*p* = 0.002). Similar findings have been reported in other RCC studies, where *NODAL* overexpression promotes cell proliferation, invasion, and suppression of RCC cell deaths (*p* < 0.05) [60]. Wu et al. reported that high expression of *NODAL* in RCC patients is associated with angiogenesis, including vasculogenic mimicry, and is linked to shorter OS (*p* = 4.652 × 10^−5^) and DFS (*p* = 1.202 × 10^−4^) [61].

Mutations in *ITM2B* and *SYT5* were observed in the Korean-KIRP dataset at frequencies of 1.67% and 3.33%, respectively. These mutations were found to be significantly associated with DFS (*p* = 0.027 and *p* = 0.021, respectively). However, there is currently no reported association between mutations in these genes and the survival of cancer patients.

Meanwhile, among 293 patients in the TCGA-KIRP cohort, 7 out of 12 patients with metastases died. Uniquely, only patients who died had mutations in survival-specific genes found in this study (*CIITA*, *ESRP2*, *MAOB*, *MUC17*, *MYH10*, *PPM1F*, *RTL1*, *RYR1*, *SNX7*, *SSX2IP*, and *THUMPD2*). In particular, the frequency of metastasis was higher in patients with mutations in the *ESRP2* (50%), *MUC17* (33%), *SNX7* (50%), and *SSX2IP* (100%) genes. Therefore, we would like to focus our discussion on genes that have been reported in cancer research, such as ESRP2, MUC17, MYH10, and SSX2IP.

Epithelial splicing regulatory protein 2 (*ESRP2*) belongs to the heterogeneous nuclear ribonucleoprotein family of RNA binding proteins, playing a pivotal role in regulating the alternative splicing events of pre-mRNAs [26]. Miyazono et al. indicated a potential correlation between the function of *ESRP2* and the prognosis of ccRCC patients [26]. Their findings propose that Arkadia plays a critical role in suppressing the progression of ccRCC by regulating *ESRP2* function. They proposed that the Arkadia–ESRP2 axis could act as a tumor suppressor in ccRCC [26]. Association of *ESRP2* with metastasis has been reported in RCC. Similar associations with metastasis have been reported in other cancers as well. A study on RCC metastasis reported that the stability of *ESRP2* regulated by the METTL14-mediated Lnc-LSG1 m6A modification plays a crucial role in the metastasis and progression of ccRCC [62]. Additionally, a laryngeal cancer research study has reported a negative correlation between *ESRP2* expression and lymphatic metastasis [63]. High *ESRP2* expression associated with lymph node metastasis (*p* < 0.0001) has been reported in prostate cancer [64]. In bladder cancer, a high level of *ESRP2* expression has been linked to lung metastasis (*p* < 0.0001) [65].

Mucin 17, cell surface associated (MUC17) was recently reported in RCC as one of a diverse set of mucin gene variants; many variants of MUC17 have been reported to exist [66]. It is associated with shorter overall survival and a trend of shorter progression-free survival in RCC patients, implying a potential role for mucin gene variants in influencing prognosis and treatment response. In our study, among the 7 patients out of 12 who died with metastases in the TCGA cohort, 3 patients (25.0%) had MUC17 mutations. Alterations in mucin family genes are common in ccRCC, and it is noteworthy that recent studies have highlighted the importance of mucin, which is frequently overexpressed in various epithelial malignancies, including RCC [67]. Mucins such as MUC17 and MUC1 have been reported in relevant studies in renal cell carcinoma [67], although MUC1 was not identified as a survival-specific gene in our study.

Myosin Heavy Chain 10 (*MYH10*) regulates the positioning of Golgi apparatus, influencing cell movement [68]. There is currently no reported research on *MYH10* in RCC. However, a hepatocellular carcinoma (HCC) research study has reported that genomic depletion at the *MYH10* locus (17p13.1) is significantly correlated with decreased OS (*p* = 0.017, HR = 1.55) and DFS (*p* = 0.003, HR = 1.59) and that the downregulation of *MYH10* can promote HCC metastasis [68]. A serous ovarian cancer research study has reported that *MYH10* expression is associated with intraperitoneal metastasis (*p* < 0.0001) and intestinal metastasis (*p* = 0.0281) [69].

The Synovial sarcoma X breakpoint 2 interacting protein (*SSX2IP*) gene encodes a protein that is a component of the cell adhesion system, functioning at adherens junctions to connect the actin cytoskeleton to the plasma membrane and mediate cell signaling and adhesion [70]. While there are no reports on *SSX2IP* mutations in RCC, a study on nasopharyngeal carcinoma has revealed that *SSX2IP* overexpression is associated with disease specific survival (hazard ratio: 4.290; 95% confidence interval: 2.271–8.102; *p* < 0.001) and distant metastasis-free survival (HR: 4.159; 95% CI: 2.072–8.345; *p* < 0.001) [70]. A hepatocellular carcinoma research study has reported that the *SSX2IP* gene can induce the occurrence and metastasis of cancer cells and that the survival time in the *SSX2IP* high expression group is shorter than that in the low expression group (*p* = 0.004) of 51 HCC patients [71]. 

Among eight patients with metastasis in the 60-patient Korean-KIRP cohort, mutations in 10 survival gene signatures were observed in six patients. Notably, genes identified in these metastatic patients—*BAP1*, *BRAF*, and *EGFR*—have been previously reported in association with cancer metastasis [72,73,74]. In one (1.67%) patient with metastatic kidney cancer in the Korean-KIRP dataset, it was particularly noteworthy that mutations were identified in all three genes, *BAP1*, *BRAF*, and *EGFR*. This patient was a 51-year-old male diagnosed with Type 2 PRCC, classified as T2N0M1 according to TNM staging. Recurrence occurred three months after surgery, and the patient died 44 months after the initial diagnosis. Previous PRCC studies have not yet reported an association of prognosis with *BAP1*, *BRAF*, and *EGFR* mutations. 

There have been reports of differences in gene mutations according to histological type of PRCC [1,75]. Linehan et al. conducted a comprehensive molecular analysis of 161 PRCC tumors, revealing a high association of Type 1 tumors with mutations in the *MET* proto-oncogene and *EGFR*, while Type 2 tumors display greater heterogeneity, with frequent mutations in *CDKN2A*, *SETD2*, and *TFE3* fusions [1,75]. In this study, the analysis of gene mutations in Korean patients based on Type I and Type II PRCC revealed that eight survival-specific genes were observed only in Type II PRCC. These genes included *BAP1*, *EGFR*, *ITM2B*, *JAK1*, *NODAL*, *PCSK2*, *SPATA13*, and *SYT5*, with mutation frequencies of 2.22%, 6.67%, 2.22%, 6.67%, 2.22%, 2.22%, 20.00%, and 4.44%, respectively. In comparison with Type II, genes with higher mutation frequencies in Type I were *BRAF* (Type I: 6.67% vs. Type II: 4.44%) and *CFDP1* (Type I: 26.67% vs. Type II: 11.11%).

To explore the mechanism of the survival gene signatures in PRCC, we performed GSEA and identified gene ontology and KEGG pathways, such as primary amine oxidase activity (*p* = 0.00509), histone reader activity (*p* = 0.006781), and tRNA methyltransferase activity (*p* = 0.009313) in 17 survival-specific genes of th eTCGA-KIRP dataset, and protein tyrosine kinase activity (*p* = 0.0167), phosphatase binding (*p* = 0.0167), and ubiquitin protein ligase binding (*p* = 0.03922) in 10 survival-specific genes of the Korean-KIRP dataset. Various studies on the molecular features of RCC have yielded important new insights into the molecular drivers underlying ccRCC ontogeny and progression [76]. For example, because *PTEN* mutations are associated with a poor prognosis in ccRCC, targeted treatment with selective AKT inhibitors has been proposed for patients with mutations [76].

Identified gene mutations associated with survival in PRCC patients generally had very low occurrence frequencies. There were significant variations in types of gene mutations depending on the database. Specifically, in the TCGA-KIRP dataset, patients with mutations in the survival-specific genes *ESRP2* (50%), *MUC17* (33%), *SNX7* (50%), and *SSX2IP* (100%) showed a higher frequency of metastasis than those without such mutations. In the Korean-KIRP dataset, patients with mutations in *BAP1* (100%), *ITM2B* (100%), *NODAL* (100%), *PCSK2* (100%), and *JAK1* (67%) also exhibited a higher frequency of metastasis than those without such mutations. This suggests an association between the occurrence of metastasis and the presence of mutations in certain survival-specific genes.

The present study has several limitations. First, while the TCGA-KIRP dataset included 293 patients, the Korean-KIRP cohort comprised only 60 patients, which was relatively small. Second, the low frequency of genetic mutations in survival-specific genes in the TCGA-KIRP dataset (ranging from 0.34% to 3.07%) raises concerns about statistical robustness. Third, this study did not account for potential racial differences between the two groups. Considering these limitations, a direct comparison of results between the two cohorts may not be appropriate. Future studies with a larger sample size of PRCC patients are needed to obtain more accurate results. Fourth, discussion of our findings is somewhat limited because many of the key genes we discovered have not previously been reported for survival specificity in RCC. Most of the cases where relevant data were available were based on ccRCC rather than on PRCC.

In summary, we used machine learning to derive genes associated with survival in PRCC, and next generation sequencing was conducted on 60 Korean PRCC patients using a customized PRCC gene panel consisting of 202 genes, including 171 survival-specific genes. Identified genetic variations were statistically validated. The survival-specific genes identified in this study comprised 40 gene signatures from TCGA-KIRP and 10 gene signatures from Korean PRCC patients. Notably, *PCSK2* was identified as the only survival-specific gene in both the TCGA-KIRP and Korean-KIRP databases. Patients with mutations in each of these 40 genes consistently showed lower survival rates than those without such mutations. The collective survival analysis for 17 survival-specific genes demonstrated that patients with mutations in any of these genes had significantly lower survival compared to patients without mutations in both OS and DFS. Of 10 gene signatures from Korean PRCC patients, most patients with gene mutations had shorter survival rates than those without mutations, except for the *ITM2B*, *JAK1*, *NODAL*, and *SYT5* genes. The collective survival analysis for three genes (*BAP1*, *PCSK2*, and *SPATA13*) demonstrated survival specificity in both OS and DFS, which was significantly lower in patients with mutations.

In conclusion, we identified and validated genes specific for the survival of PRCC patients in TCGA and Korean databases. In particular, the survival gene signature, including PCSK2 commonly obtained from the 40 gene signature of TCGA and the 10 gene signature of the Korean database, is expected to provide insight into predicting the survival of PRCC patients and developing new treatment modalities.

## Figures and Tables

**Figure 1 cancers-16-02006-f001:**
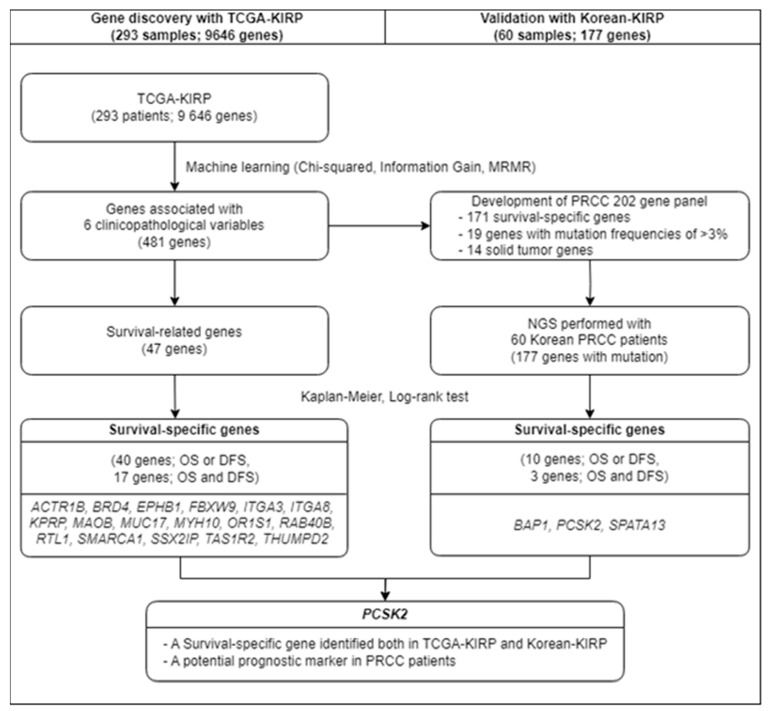
Workflow of the study.

**Figure 2 cancers-16-02006-f002:**
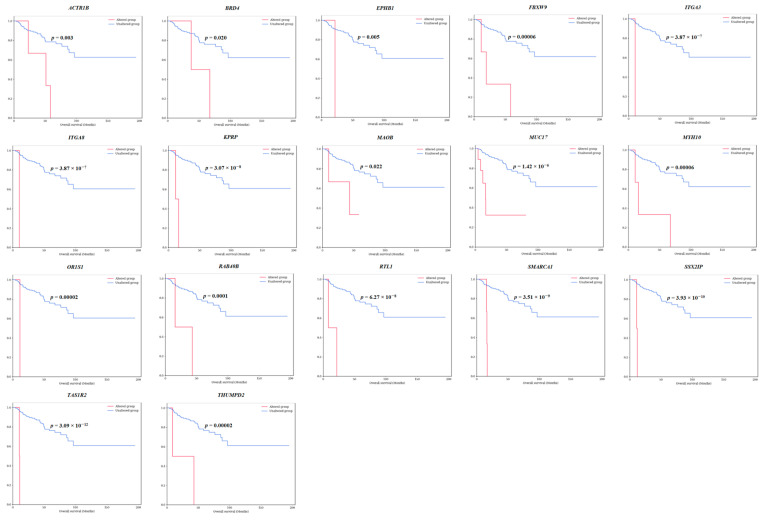
Overall survival analysis of 17 survival-specific genes identified in TCGA-KIRP dataset (*ACTR1B*, *BRD4*, *EPHB1*, *FBXW9*, *ITGA3*, *ITGA8*, *KPRP*, *MAOB*, *MUC17*, *MYH10*, *OR1S1*, *RAB40B*, *RTL1*, *SMARCA1*, *SSX2IP*, *TAS1R2*, and *THUMPD2*). The *p*-values are from the Log-rank test.

**Figure 3 cancers-16-02006-f003:**
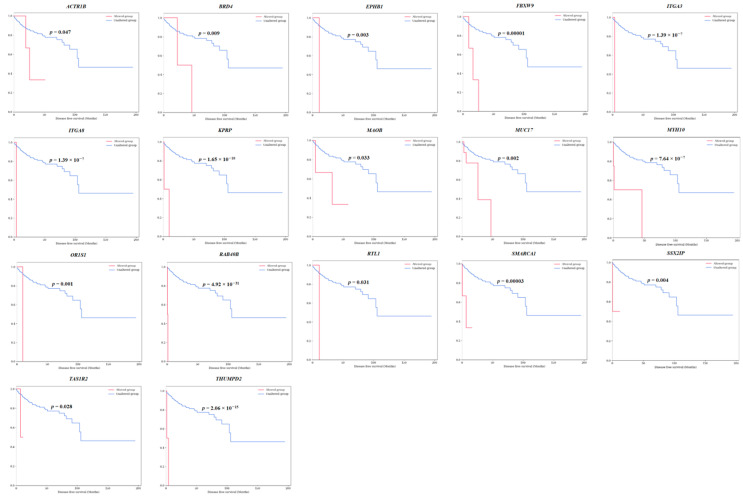
Disease-free survival analysis of 17 survival-specific genes identified in TCGA-KIRP dataset (*ACTR1B*, *BRD4*, *EPHB1*, *FBXW9*, *ITGA3*, *ITGA8*, *KPRP*, *MAOB*, *MUC17*, *MYH10*, *OR1S1*, *RAB40B*, *RTL1*, *SMARCA1*, *SSX2IP*, *TAS1R2*, and *THUMPD2*). The *p*-values are from the Log-rank test.

**Figure 4 cancers-16-02006-f004:**
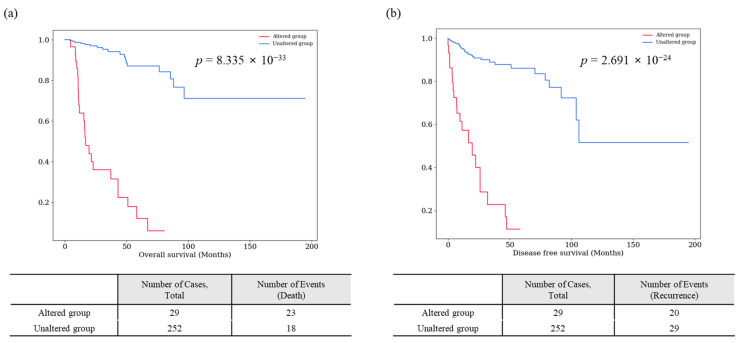
Collective survival analysis of 17 survival-specific genes identified in TCGA-KIRP dataset (*ACTR1B*, *BRD4*, *EPHB1*, *FBXW9*, *ITGA3*, *ITGA8*, *KPRP*, *MAOB*, *MUC17*, *MYH10*, *OR1S1*, *RAB40B*, *RTL1*, *SMARCA1*, *SSX2IP*, *TAS1R2*, and *THUMPD2).* Overall survival (**a**) and disease-free survival (**b**) graphs are shown. The *p*-values are from the Log-rank test.

**Figure 5 cancers-16-02006-f005:**
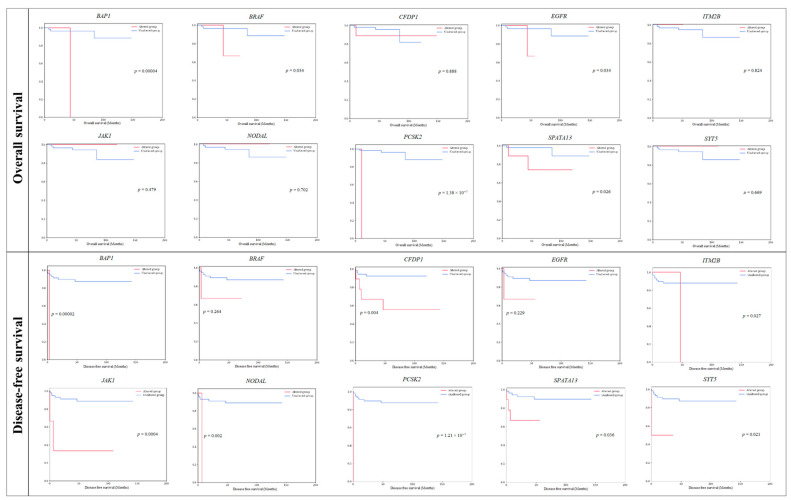
Survival analysis of 60 Korean-KIRP patients for 10 survival-specific genes (*BAP1*, *BRAF*, *CFDP1*, *EGFR*, *ITM2B*, *JAK1*, *NODAL*, *PCSK2*, *SPATA13*, and *SYT5).* Overall survival and disease-free survival graphs are shown. The *p*-values are from the Log-rank test.

**Figure 6 cancers-16-02006-f006:**
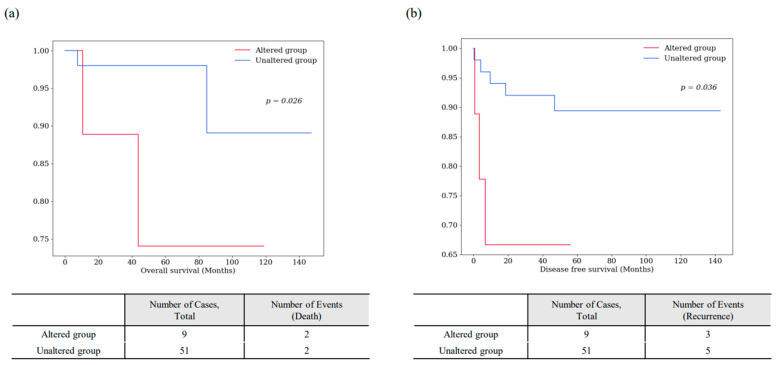
Collective survival analysis of three survival-specific genes identified in Korean-KIRP (*BAP1*, *PCSK2*, and *SPATA13*). Graphs of overall survival (**a**) and disease-free survival (**b**) are shown. The *p*-values are from the Log-rank test.

**Figure 7 cancers-16-02006-f007:**
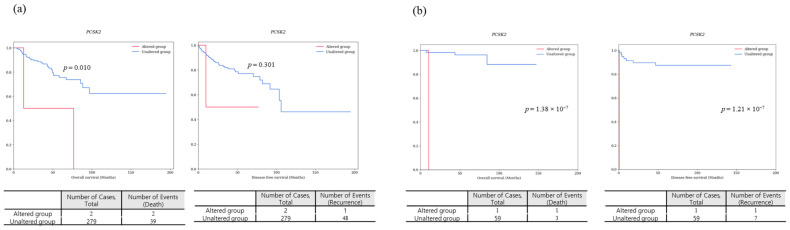
Survival graphs of *PCSK2* gene comparing survival rates based on TCGA-KIRP (**a**) and Korean-KIRP (**b**) databases. The *p*-values are from the Log-rank test.

**Table 1 cancers-16-02006-t001:** Clinical characteristics of 293 patients in the TCGA-KIRP database.

Variables		Number of Patients (%), *N* = 293
Sex	Male	214 (73.0)
	Female	78 (26.6)
	Not available	1 (0.4)
Survival	Alive	248 (84.6)
	Deceased	44 (15.0)
	Not available	1 (0.4)
Recurrence	Disease free	218 (74.4)
	Recurred/Progressed	54 (18.4)
	Not available	21 (7.2)
Metastasis	Absent	209 (71.3)
	Present	12 (4.1)
	Not available	72 (24.6)

TCGA-KIRP, Cancer Genome Atlas-Kidney Renal Papillary Cell Carcinoma dataset.

**Table 2 cancers-16-02006-t002:** A total of 40 survival-specific genes discovered from 293 TCGA-KIRP patients and findings of patients with mutations.

No.	Gene	Number of Patients with Mutation (%)	Cytoband	Mutation Type				Survival (%)		Metastasis (%)			Overall Survival	Disease Free Survival
				Truncating	Missense	Splice	Inframe	Alive	Deceased	Absent	Present	Not Available		
1	*ACTR1B*	3 (1.02)	2q11.2	1	2	0	0	0	3 (100)	2 (67)	0	1 (33)	**0.003 ***	**0.047 ***
2	*BPNT1*	2 (0.68)	1q41	1	1	1	0	0	2 (100)	1 (50)	0	1 (50)	**0.011 ***	0.204
3	*BRD4*	2 (0.68)	19p13.12	1	1	0	0	0	2 (100)	2 (100)	0	0	**0.020 ***	**0.009 ***
4	*BZRAP1*	2 (0.68)	17q22	0	2	0	0	0	2 (100)	2 (100)	0	0	**0.003 ***	0.512
5	*C15orf27*	1 (0.34)	15q24.2	0	1	0	0	0	1 (100)	1 (100)	0	0	**0.019 ***	0.652
6	*C16orf72*	2 (0.68)	16p13.2	0	2	0	0	0	2 (100)	1 (50)	0	1 (50)	**0.002 ***	0.144
7	*CD1C*	2 (0.68)	1q23.1	1	1	0	0	0	2 (100)	1 (50)	0	1 (50)	**0.017 ***	0.410
8	*CEP128*	2 (0.68)	14q31.1	2	0	0	0	0	2 (100)	1 (50)	0	1 (50)	**0.017 ***	0.330
9	*CIITA*	2 (0.68)	16p13.13	0	2	0	0	0	2 (100)	1 (50)	1 (50)	0	**0.000002 ****	0.140
10	*COL5A1*	4 (1.37)	9q34.3	2	1	1	0	1 (25)	3 (75)	3 (75)	0	1 (25)	**0.001 ****	0.404
11	*CYP51A1*	2 (0.68)	9q34.3	0	2	0	0	0	2 (100)	0	0	2 (100)	**0.001 ***	0.139
12	*DNAAF2*	2 (0.68)	14q21.3	1	1	0	0	0	2 (100)	2 (100)	0	0	**0.001 ***	0.548
13	*EPHB1*	1 (0.34)	3q22.2	0	1	0	0	0	1 (100)	1 (100)	0	0	**0.005 ***	**0.003 ***
14	*ESRP2*	2 (0.68)	16q22.1	1	1	0	0	0	2 (100)	0	1 (50)	1 (50)	**0.016 ***	0.492
15	*FBXW9*	3 (1.02)	19p13.13	0	3	0	0	0	3 (100)	3 (100)	0	0	**0.0001 ****	**0.00001 ****
16	*ITGA3*	1 (0.34)	17q21.33	0	1	0	0	0	1 (100)	0	0	1 (100)	**3.87 × 10^−7^ ****	**1.42 × 10^−7^ ****
17	*ITGA4*	1 (0.34)	2q31.3	0	1	0	0	0	1 (100)	1 (100)	0	0	**0.00001 ****	0.766
18	*ITGA8*	1 (0.34)	10p13	0	1	0	0	0	1 (100)	0	0	1 (100)	**3.87 × 10^−7^ ****	**1.42 × 10^−7^ ****
19	*KIF5C*	2 (0.68)	2q23.1-q23.2	1	1	0	0	0	2 (100)	1 (50)	0	1 (50)	**0.0001 ****	0.146
20	*KPRP*	2 (0.68)	1q21.3	0	2	0	0	0	2 (100)	2 (100)	0	0	**3.07 × 10^−8^ ****	**1.68 × 10^−10^ ****
21	*MAOB*	3 (1.02)	Xp11.3	0	2	1	0	1 (33)	2 (67)	2 (67)	1 (33)	0	**0.022 ***	**0.033 ***
22	*MUC17*	9 (3.07)	7q22.1	1	9	0	1	4 (44)	5 (56)	1 (11)	3 (33)	5 (56)	**1.42 × 10^−6^ ****	**0.002 ***
23	*MYH10*	4 (1.37)	17p13.1	1	3	0	0	1 (25)	3 (75)	2 (50)	1 (25)	1 (25)	**0.00006 ****	**6.40 × 10^−7^ ****
24	*OGFR*	1 (0.34)	20q13.33	0	1	0	0	0	1 (100)	1 (100)	0	0	**0.0006 ****	0.736
25	*OR1S1*	1 (0.34)	11q12.1	0	1	0	0	0	1 (100)	1 (100)	0	0	**0.00002 ****	**0.0007 ****
26	*PCBP4*	2 (0.68)	3p21.2	0	2	0	0	0	2 (100)	1 (50)	0	1 (50)	**0.025 ***	0.481
27	*PCGF2*	2 (0.68)	17q12	1	1	0	0	0	2 (100)	1 (50)	0	1 (50)	**0.042 ***	0.441
28	*PCSK2*	2 (0.68)	20p12.1	0	1	0	1	0	2 (100)	2 (100)	0	0	**0.010 ***	0.301
29	*PLEKHB2*	2 (0.68)	2q21.1	1	2	0	0	0	2 (100)	2 (100)	0	0	**0.002 ***	0.175
30	*PPM1F*	2 (0.68)	22q11.22	1	1	0	0	0	2 (100)	1 (50)	1 (50)	0	**0.0002 ****	0.180
31	*RAB40B*	2 (0.68)	17q25.3	0	2	0	0	0	2 (100)	1 (50)	0	1 (50)	**0.0001 ****	**5.88 × 10^−31^ ****
32	*RRP36*	2 (0.68)	6p21.1	1	1	0	0	0	2 (100)	2 (100)	0	0	**0.006 ***	0.176
33	*RTL1*	2 (0.68)	14q32.2	3	0	0	1	0	2 (100)	1 (50)	1 (50)	0	**6.27 × 10^−8^ ****	**0.031 ***
34	*RYR1*	7 (2.39)	19q13.2	1	6	0	0	3 (43)	4 (57)	6 (86)	1 (14)	0	**0.001 ****	0.051
35	*SMARCA1*	3 (1.02)	Xq25-q26.1	1	2	0	0	0	3 (100)	1 (33)	0	2 (67)	**3.51 × 10^−9^ ****	**2.64 × 10^−5^ ****
36	*SNX7*	2 (0.68)	1p21.3	1	1	0	0	0	2 (100)	0	1 (50)	1 (50)	**0.0004 ****	0.197
37	*SSX2IP*	2 (0.68)	1p22.3	0	2	0	0	0	2 (100)	0	2 (100)	0	**3.93 × 10^−10^ ****	**0.006 ***
38	*TAS1R2*	2 (0.68)	1p36.13	0	2	0	0	0	2 (100)	2 (100)	0	0	**3.09 × 10^−12^ ****	**0.028 ***
39	*THUMPD2*	2 (0.68)	2p22.1; 2p22-p21	2	0	1	0	0	2 (100)	1 (50)	1 (50)	0	**0.00002 ****	**2.15 × 10^−15^ ****
40	*VPS13D*	2 (0.68)	1p36.22-p36.21	0	2	0	0	0	2 (100)	1 (50)	0	1 (50)	**0.042 ***	0.441

Statistically significant values are bolded. * is used for *p*-values < 0.05. ** are used for *p*-values < 0.001. The *p*-values are from the Log-rank test. TCGA-KIRP, Cancer Genome Atlas-Kidney Renal Papillary Cell Carcinoma dataset.

**Table 3 cancers-16-02006-t003:** Analysis of 12 patients who developed distant metastases among 293 patients in the TCGA-KIRP database.

Patients with Metastasis	Age	Sex	TNM Stage			Survival	Tumor Type	Survival-Specific Genes		
			T	N	M					
TCGA-B9-4114-01	49	M	2	0	1	Alive	NA			
TCGA-G7-A8LB-01	70	M	2	NA	1	Alive	NA			
TCGA-BQ-5894-01	42	M	3	1	1	Alive	II			
TCGA-4A-A93X-01	58	M	3	1	1	Alive	NA			
TCGA-F9-A8NY-01	38	F	4	1	1	Alive	II			
TCGA-A4-A57E-01	59	M	2	0	1	Deceased	NA	*MUC17*	*RTL1*	*RYR1*
TCGA-2Z-A9J7-01	63	M	2	0	1	Deceased	NA	*MUC17*		
TCGA-AL-3466-01	41	M	3	1	1	Deceased	NA	*MAOB*		
TCGA-BQ-5877-01	60	M	3	1	1	Deceased	NA	*PPM1F*	*THUMPD2*	
TCGA-SX-A7SM-01	60	M	3	1	1	Deceased	II	*ESRP2*	*MUC17*	*SSX2IP*
TCGA-BQ-5893-01	61	M	3	1	1	Deceased	NA	*CIITA*	*SNX7*	
TCGA-BQ-5889-01	63	M	3	1	1	Deceased	NA	*MYH10*	*SSX2IP*	

**Table 4 cancers-16-02006-t004:** Clinicopathologic findings of 60 patients in the Korean-KIRP database.

Variables			Patients (%) *N* = 60	Survival (%)		*p*-Value
				Alive	Deceased	
Age	<70	<50	11 (18.3)	9 (24.3)	2 (5.4)	1.000
		50–59	11 (18.3)	10 (27.0)	1 (2.7)	
		60–69	15 (25)	15 (40.5)	0 (0)	
	≥70	70–79	15 (25)	15 (65.2)	0 (0)	
		80–89	7 (11.7)	7 (30.4)	0 (0)	
		≥90	1 (1.7)	0 (0)	1 (4.3)	
Sex	Male		45 (75)	42 (93.3)	3 (6.7)	1.000
	Female		15 (25)	14 (93.3)	1 (6.7)	
Tumor type	I		15 (25)	15 (100.0)	0 (0)	0.564
	II		45 (75)	41 (91.1)	4 (8.9)	
Nuclear Grade	I		0 (0)	0 (0)	0 (0)	0.133
	II		25 (41.7)	25 (100.0)	0 (0)	
	III		34 (56.7)	31 (88.6)	3 (8.6)	
	IV		1 (1.7)	0 (0)	1 (2.9)	
Tumor size	≤7.0 cm		52 (86.7)	51 (98.1)	1 (1.9)	0.006
	>7.0 cm		8 (13.3)	5 (62.5)	3 (37.5)	
T stage	T1		47 (78.3)	46 (97.9)	1 (2.1)	0.029
	T2		5 (8.3)	3 (23.1)	2 (15.4)	
	T3		8 (13.3)	7 (53.8)	1 (7.7)	
N stage	N0		57 (95)	54 (94.7)	3 (5.3)	0.190
	N1		3 (5)	2 (66.7)	1 (33.3)	
M stage	M0		52 (86.7)	52 (100.0)	0 (0)	0.0001
	M1		8 (13.3)	4 (50.0)	4 (50.0)	
Recurrence	Disease free		52 (86.7)	52 (100.0)	0 (0)	0.0001
	Recurred/Progressed		8 (13.3)	4 (50.0)	4 (50.0)	
Response to laparoscopic	No evidence of disease		52 (86.7)	52 (100.0)	0 (0)	0.0001
radical nephrectomy	Fail		8 (13.3)	4 (50.0)	4 (50.0)	

KIRP, kidney renal papillary cell carcinoma. The *p*-values are from the Fisher’s exact test.

**Table 5 cancers-16-02006-t005:** A total of 10 survival-specific genes in 60 patients of the Korean-KIRP database and findings of patients with mutations.

No.	Gene	Number of Patients with Mutation	Mutation Frequency, %	Cytoband	Mutation Type			Survival (%)		Metastasis (%)		Overall Survival	Disease Free Survival
					Truncating	Missense	Inframe	Alive	Deceased	Absent	Present		
1	*BAP1*	1	1.67	3p21.1	0	0	1	0	1 (100)	0	1 (100)	**0.00004 ****	**0.00002 ****
2	*BRAF*	3	5.00	7q34	0	3	5	2 (67)	1 (33)	2 (67)	1 (33)	**0.034 ***	0.264
3	*CFDP1*	9	15.00	16q23.1	0	15	1	8 (89)	1 (11)	5 (56)	4 (44)	0.888	**0.004 ***
4	*EGFR*	3	5.00	7p11.2	0	4	0	2 (67)	1 (33)	2 (67)	1 (33)	**0.034 ***	0.229
5	*ITM2B*	1	1.67	13q14.2	0	1	0	1 (100)	0	0	1 (100)	0.814	**0.027 ***
6	*JAK1*	3	5.00	1p31.3	1	4	1	3 (100)	0	1 (33)	2 (67)	0.479	**0.0004 ****
7	*NODAL*	1	1.67	10q22.1	0	1	0	1 (100)	0	0	1 (100)	0.702	**0.002 ***
8	*PCSK2*	1	1.67	20p12.1	0	1	0	0	1 (100)	0	1 (100)	**1.38 × 10^−7^ ****	**1.21 × 10^−7^ ****
9	*SPATA13*	9	15.00	13q12.12	0	9	0	7 (78)	2 (22)	6 (67)	3 (33)	**0.026 ***	**0.036 ***
10	*SYT5*	2	3.33	19q13.42	0	2	0	2 (100)	0	1 (50)	1 (50)	0.669	**0.021 ***

Statistically significant values are bolded. * is used for *p-*values < 0.05. ** are used for *p*-values < 0.001. The *p*-values are from the Log-rank test. KIRP, kidney renal papillary cell carcinoma.

**Table 6 cancers-16-02006-t006:** Analysis of 8 patients who developed distant metastases among 60 patients in the Korean-KIRP database.

Patients with Metastasis	Age	Sex	Tumor Type	Nuclear Grade	TNM Stage			Survival	Metastatic Site	Response to LRN	Mutations among 10 Survival-Specific Genes			
					T	N	M							
S_20220322_036	80	M	Ⅱ	III	3	0	1	Alive	Lung	Fail	*CFDP1*	*ITM2B*		
S_20220425_038	42	F	Ⅱ	III	1	0	1	Alive	Adrenal gland, Lung	Fail	*JAK1*	*SYT5*		
S_20220425_044	77	M	Ⅱ	III	3	0	1	Alive	Bone	Fail	*CFDP1*	*JAK1*	*NODAL*	*SPATA13*
S_20220425_055	82	F	Ⅱ	III	3	1	1	Alive	Lymph node	Fail	*CFDP1*			
S_20220425_042	49	M	Ⅱ	III	2	0	1	Deceased	Lung, Lymph node	Fail				
S_20220322_058	42	M	Ⅱ	III	1	0	1	Deceased	Lung	Fail				
S_20220425_029	51	M	Ⅱ	III	2	0	1	Deceased	Lung	Fail	*BAP1*	*BRAF*	*EGFR*	*SPATA13*
S_20220425_040	93	F	Ⅱ	IV	3	1	1	Deceased	Liver, Lung, Vagina	Fail	*CFDP1*	*PCSK2*	*SPATA13*	

LRN, laparoscopic radical nephrectomy.

**Table 7 cancers-16-02006-t007:** Comparison of a survival-specific gene commonly identified in both TCGA-KIRP and Korean-KIRP databases.

Gene	Database	Mutation				Survival Analysis	
		Type	HGVS.c	HGVS.p	Frequency, %	Overall Survival	Disease Free Survival
*PCSK2*	TCGA-KIRP	missense_variant	c.850C>T	p.Leu284Phe	0.68	**0.010 ***	0.301
		In_Frame_Ins	c.10_12dupGGT	p.Gly4dup			
	Korean-KIRP	missense_variant	c.1879G>T	p.Val627Leu	1.67	**1.38** **× 10^−7^ ****	**1.21** **× 10^−7^ ****

Statistically significant values are bolded. * is used for *p*-values < 0.05. ** are used for *p*-values < 0.001. The *p*-values are from the Log-rank test. KIRP, kidney renal papillary cell carcinoma; TCGA-KIRP, Cancer Genome Atlas-Kidney Renal Papillary Cell Carcinoma dataset.

**Table 8 cancers-16-02006-t008:** Three out of ten survival-specific genes in the Korean-KIRP cohort were correlated with clinicopathologic factors.

Clinical Variable	Statistic	*CFDP1*	*JAK1*	*SPATA13*
Tumor size	OR (CI)	0.47 (0.06–5.71)	0.29 (0.01–18.86)	0.11 (0.02–0.80)
	*p*-value	0.593	0.354	**0.013 ***
T stage	OR (CI)	0.28 (0.05–1.67)	0.54 (0.03–34.06)	0.16 (0.02–0.89)
	*p*-value	0.092	0.526	**0.018 ***
N stage	OR (CI)	13.20 (0.62–852.80)	0.00 (0.00–60.70)	2.98 (0.05–63.96)
	*p*-value	0.056	1.000	0.391
M stage	OR (CI)	8.83 (1.25–66.35)	15.50 (0.71–1012.77)	4.44 (0.55–30.90)
	*p*-value	**0.013 ***	**0.044 ***	0.090
Recurrence	OR (CI)	8.83 (1.25–66.35)	15.50 (0.71–1012.77)	4.44 (0.55–30.90)
	*p*-value	**0.013 ***	**0.044 ***	0.090
Death	OR (CI)	1.97 (0.03–28.41)	0.00 (0.00–40.77)	6.64 (0.42–105.71)
	*p*-value	0.488	1.000	0.103
Response to LRN	OR (CI)	0.11 (0.02–0.80)	0.06 (0.00–1.40)	0.23 (0.03–1.81)
	*p*-value	**0.013 ***	**0.044 ***	0.090

Statistically significant values are bolded. * is used for *p*-values < 0.05. The *p*-values are from the Fisher’s exact test. CI, confidence interval; OR, odds ratio; LRN, laparoscopic radical nephrectomy.

## Data Availability

The data from the TCGA-KIRP and Korean-KIRP databases can be found in the Appendix A.

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
