# Peer review of "Discovery and Validation of Survival-Specific Genes in Papillary Renal Cell Carcinoma Using a Customized Next-Generation Sequencing Gene Panel"

_cancers, 2024, doi:10.3390/cancers16112006_

Round 1

Reviewer 1 Report

Comments and Suggestions for Authors

The study is particularly interesting as it addresses a significant gap in the understanding of papillary renal cell carcinoma, specifically its genetic underpinnings related to patient survival. The novelty of the study lies in the use of a customized next-generation sequencing gene panel to identify and validate survival-specific genes in PRCC patients from two distinct databases, TCGA-KIRP and a Korean cohort (Korean-KIRP). The validation of genes such as PCSK2 across different ethnic groups adds to the uniqueness and relevance of the research, suggesting potential universal biomarkers for PRCC prognosis.

Areas for Improvement:

  • Discussion on Contradictory Findings: There is little discussion about why certain genes identified in previous studies did not show survival specificity in this study. Addressing these discrepancies would enrich the discussion.
  • Article Readability:It's crucial to make the text clearer and more direct. Currently, there are many complex sentences and an excessive use of technical jargon that can make reading difficult. Try simplifying the language where possible and breaking up longer sentences into smaller segments to enhance overall comprehension.
  • Organization of Graphic Materials:Graphic materials, such as figures and tables, are essential for illustrating data and supporting your arguments. However, their current arrangement might be somewhat confusing. Ensure that each graphic element is clearly linked to the text that references it, and that legends are concise and informative. Consider reorganizing or redesigning some graphics for better integration into the document's flow.
  • Enhancing the Discussion:The discussion section is crucial for emphasizing the importance and implications of your findings. It could currently benefit from a more engaging and critical approach. Discuss in more depth how your results compare with existing publications and what new value they add to the field. You might also expand on how these findings could influence future research or clinical practice, making the discussion not just more informative but also more stimulating and compelling for the reader. 
  • Recent studies, including those documented in the article with PMID: 37685983, have highlighted the importance of cancer stem cell signaling pathways such as Notch, Wnt, and Hedgehog in clear cell renal cell carcinoma, suggesting that these pathways may also be crucial in other renal carcinoma subtypes such as papillary renal cell carcinoma. This knowledge could expand our understanding of tumorigenicity and genetic variation in renal tumors, complementing our current efforts to identify survival-specific biomarkers in PRCC using next-generation sequencing panels. Please cite and discuss this.
  • In addition, recent research has underscored the importance of MUC1, a mucin frequently overexpressed in various epithelial malignancies including renal carcinoma types (PMID: 38540735). This work reviews the role of MUC1 in cancer cell proliferation, metabolic reprogramming, chemoresistance, and angiogenesis. Considering our discovery of genetic markers in papillary renal cell carcinoma, exploring the overexpression of MUC1 in our context could provide deeper insights into the interactions between genetic pathways and pathological processes in RCC, potentially guiding the development of targeted therapies. Please cite and discuss it in your paper.
  • check typos 

Reviewer 2 Report

Comments and Suggestions for Authors

The authors should be congratulated for their work and for addressing an important topic such as the discovery and validation of survival-specific genes in papillary RCC. Only a few points warrant mention:

Major comments:

1.    In the “Discussion” section, the authors correctly present a brief summary of results and data from the literature. However, in this section, they miss a real debate with literature data. Indeed, previous findings are based mostly on ccRCC and not on pRCC, thus a more extended discussion of current findings should be recommendable. Similarly, I would ask to the authors to better elucidate the clinical implication of these findings.

Moreover, the conclusions are too short.

Minor comments:

1.    In the “Introduction” section, to better highlight the aim of the study, the authors should briefly show data on survival, as well as therapeutic options. Moreover, some biomarkers for RCC diagnosis have already been discovered, as such see PMID: 37446024.

2.    In the “Material and Methods” section, which “both somatic non-silent mutation” have been selected? I suggest better to clarify this process of selection in this section.

3.    In the “Material and Methods” section, “2.11. Statistical Analysis” subsection, why did the authors choose 70 years old as a threshold to divide the population?

4.    Line 433, “ccrcc”, please fix the typo.

5.    In the whole manuscript the authors report that some genes are related to “survival” in RCC. However, I find this term misunderstandable, as survival rates in cancers are related to several factors, not only tumor-specific ones, could it be “higher rates of survival”?

Round 2

Reviewer 2 Report

Comments and Suggestions for Authors

Dear authors thank you for addressing all my concerns. You really improved the overall quality of the manuscript.